# Effect of Substitutional Elements on the Thermodynamic and Electrochemical Properties of Mechanically Alloyed La$_{1.5}$Mg$_{0.5}$Ni$_{7-x}$M$_x$ alloys (M = Al, Mn)

**Marek Nowak** , **Mateusz Balcerzak and Mieczyslaw Jurczyk** *

Institute of Materials Science and Engineering, Poznan University of Technology, Jana Pawla II 24, 61-138 Poznan, Poland; marek.nowak@put.poznan.pl (M.N.); mateusz.balcerzak@put.poznan.pl (M.B.)
* Correspondence: mieczyslaw.jurczyk@put.poznan.pl; Tel.: +48-61-665-3508

**Abstract:** The A$_2$B$_7$-type La-Mg-Ni-M-based (M = Al, Mn) intermetallic compounds were produced by mechanical alloying and annealing. The thermodynamic and electrochemical properties of these materials were studied. The nickel substitution by aluminum and manganese in the La-Mg-Ni system improves the kinetics of hydrogen absorption. The hydrogen desorption capacity of Mn substituted compounds is improved significantly, and it reaches the value of 1.79 wt.% at 303 K when the composition is La$_{1.5}$Mg$_{0.5}$Ni$_{6.80}$Mn$_{0.20}$. On the other hand, the La$_{1.5}$Mg$_{0.5}$Ni$_{6.85}$Al$_{0.15}$ shows a much higher reversible electrochemical capacity than the La$_{1.5}$Mg$_{0.5}$Ni$_7$ materials at the 50th cycle. The electrochemical discharge capacity stability increases with the increasing value of Al and Mn up to $x$ = 0.2 and 0.3, respectively. Additionally, a reduction in the discharge capacity was measured for the Al and Mn content above $x$ = 0.25 and 0.5, respectively. From the practical aspect, only La$_{1.5}$Mg$_{0.5}$Ni$_{6.80}$Mn$_{0.20}$ has a potential in the application as a hydrogen storage material.

**Keywords:** A$_2$B$_7$-type alloys; thermodynamic properties; electrochemical properties

---

## 1. Introduction

The A$_2$B$_7$-type intermetallic compounds have attracted attention as the new generation of hydrogen storage materials [1–4]. From the scientific point of view, a lot of efforts to improve the thermodynamic and electrochemical properties of the La-Mg-Ni-type materials have been undertaken. For example, modification of their chemical composition, microstructure evolution, heat and surface treatments, etc. [5–8].

The La-Mg-Ni-based compounds with light and not so expensive elements emerged as one of the most promising negative electrode materials for the Ni-MH$_x$ batteries because of their high discharge capacity, energy density, and rate capability [2]. The crystal structures of La$_{2-x}$Mg$_x$Ni$_{7.0}$ ($x$ = 0.3–0.6), RE-Mg-Ni, La$_4$MgNi$_{19}$, La$_{0.7}$Mg$_{0.3}$Ni$_{2.8}$Co$_{0.5}$-H$_2$, and Ce$_2$Ni$_7$H$_{4.7}$ materials have been discussed [9–14]. Generally, the hydrogen storage properties of ternary La-Mg-Ni hydrides are superior to corresponding binary AB$_n$ ($n$ = 2–5) [1,5–7,15–21].

The La-Ni-type alloys can be produced by the arc or induction melting in the high purity argon gas [22]. It is important to note, that modification of the hydrogen storage properties of these phases can be achieved via mechanical alloying (MA) [7,16,19–21]. Lately, the influence of magnesium content on the properties of the La$_{2-x}$Mg$_x$Ni$_7$ system was investigated in detail [7,9,23]. The content of Mg in the (La,Mg)$_2$Ni$_7$ system influenced their final phase composition. For example, the resulting main three phases (La,Mg)$_2$Ni$_7$, (La,Mg)Ni$_3$, and LaNi$_5$ were detected, for $x$ = 0.48–0.5, $x$ = 0.6, and $x$ > 0.48 in the La$_{2-x}$Mg$_x$Ni$_7$ alloy [23]. The existence of Mg in the La$_2$Ni$_7$ alloy anticipates the pulverization of their hydrides.

Moreover, Pr, Nd, Gd are frequently used to partly substitute La, while Co, Mn, Al are used to partly substitute Ni in the La-Ni system to adjust the electrochemical properties of the $MH_x$ electrode materials [24].

The discharge capacities of the Co substituted La-Mg-Ni system was studied [25]. All the $La_{1.5}Mg_{0.5}Ni_{7-x}Co_x$ ($x = 0$, 1.2, 1.8) electrodes can be activated during free cycles and have discharge capacities above 390 mAh/g. These electrodes were prepared from the mixture of alloy and carbonyl nickel powders at the weight ratio of 1:2. Nevertheless, the cyclic stability of the hydrogen storage materials becomes worse with the increase in cobalt concentration.

Recently, the effect of cobalt content and thermal treatment on the electrochemical behavior of $La_{0.7}Mg_{0.3}Ni_{2.45-x}Co_{0.75+x}Mn_{0.1}Al_{0.2}$ ($x = 0$, 0.15, 0.3) electrodes have been reported [26]. An enhancement in the cyclic stability of the electrodes was observed in the function of both concentration of Co and annealing temperature. The values of $C_{100}/C_{max}$ were 65.5% and 80.5% ($C_{100}$ is the discharge capacity after 100th charge/discharge cycles, $C_{max}$ is the maximum discharge capacity) before annealing (as-cast) and after annealing (1173 K/8 h) for $La_{0.7}Mg_{0.3}Ni_{2.15}Co_{1.05}Mn_{0.1}Al_{0.2}$, respectively. On the other hand, when the Co amount increased from $x = 0$ to $x = 0.3$, the discharge capacity of the alloy electrodes decreased.

The Mn and Al contents on the alloys phase structures and properties of the La-Mg-Ni system were investigated, as well [27,28]. The partial substitution of Ni by Mn in $RENi_{2.6-x}Mn_xCo_{0.9}$ ($x = 0$–0.9) on their phase composition and microstructure was studied [27]. In the alloys synthesized by induction melting, the following main phases were detected: $(La,Ce)_2Ni_7$ ($Ce_2Ni_7$-type structure), $(Pr,Ce)Co_3$ ($PuNi_3$-type structure), and $(La,Pr)Ni_5$ ($CaCu_5$-type structure). The hydrogen-storage capacity reached (1.04 wt.%) for $x = 0.45$. On the other hand, the discharge capacity of the electrodes initially increased from 205 mAh/g ($x = 0.0$) to 352 mAh/g ($x = 0.45$) to finally decreasing to 307 mAh/g for $x = 0.9$.

Li et al. studied the properties of $La_{0.7}Mg_{0.3}Ni_{2.55-x}Co_{0.45}Al_x$ ($x = 0$–0.4) synthesized by casting and rapid quenching [28]. Multiphase samples, consisting of $(La,Mg)Ni_3$, $LaNi_5$, and $LaNi_2$ were formed. With an increase of the Al content in the alloy, the discharge capacities monotonously decrease, while their cycle stabilities significantly increase. Additionally, the rapid quenching process deteriorates the capacity but improves the cycle stability.

The substitution of Ni with Co, Mn, Fe, Al, and Cu in $La_2MgNi_9$ decreases the hydrogen storage capacity, but at the same time increases the hydride stability [29]. Additionally, both the discharge capacity and the high-rate dischargeability of the electrodes decrease, however, the cycling stability of the substituted compositions improves.

The effects of the partial Ni replacement by Fe, Mn, and Al on the microstructures and electrochemical properties of $La_{0.7}Mg_{0.3}Ni_{2.55-x}Co_{0.45}M_x$ (M = Fe, Mn, Al; $x = 0$, 0.1), synthesized by melt spinning, was studied by Zhang et al. [30]. The amount of the $LaNi_2$ phase formed in the samples was strongly correlated with the Al and Mn contents in synthesized compounds. Significant refinement in the as-quenched samples was observed after the substitution of Al and Fe for Ni. Finally, the rapid quenching markedly enhances the cycle stabilities of the samples.

The hydriding-dehydriding properties of hydrogen storage materials can be improved by the introduction of metastable phases and the formation of nanocrystalline structures [31]. It can be achieved through the application of a nonequilibrium processing technique, such as for example, mechanical milling/alloying.

The published reports suggest that the kinetics of hydrogen absorption and desorption in the nanostructured hydrogen storage alloys can be improved due to a large specific surface area, hence, short hydrogen diffusion pathways [31–33]. It was demonstrated that MA is a powerful method for the synthesis of hydrogen storage nanopowders [21,34].

In the present research, the effect of Al and Mn on the thermodynamic and electrochemical properties of mechanically alloyed $La_{1.5}Mg_{0.5}Ni_{7-x}M_x$ (M = Al ($0 \leq x \leq 0.25$; Mn ($0 \leq x \leq 0.5$)) intermetallic compound was studied.

## 2. Materials and Methods

The nanostructured $La_{1.5}Mg_{0.5}Ni_{7-x}M_x$ (M = Al ($0 \leq x \leq 0.25$), Mn ($0 \leq x \leq 0.5$)) compounds were synthesized by mechanical alloying in a high purity argon atmosphere (Table 1). Mechanical alloying was carried out using a 8000 SPEX mixer mill (SPEX SamplePrep, Metuchen, NJ, USA) with milling frequency of 875 Hz, employing a weight ratio of hard steel balls to powder weight ratio of 4.25:1 at ambient temperature for 48 h in a continuous mode. The following metals were used: La powders—grated from rod (Alfa Aesar, 99.9%), Mg powder (Alfa Aesar, 325 mesh, 99.8%), Ni powder (Aldrich, 5 μm, 99.99%), Al powder (Aldrich, 200 mesh, 99%), and Mn powder (Aldrich, 325 mesh, ≥99%). The elemental powders were weighed, blended, and poured into a round bottom stainless vial (35 mL) in a glove box (Labmaster 130) filled with automatically controlled argon atmosphere ($O_2 \leq 2$ ppm and $H_2O \leq 1$ ppm) to obtain the materials. A composition of starting materials mixture was based on the stoichiometry of an "ideal" reaction. However, due to oxidation of La and Mg, the content of these elements was increased by 8 wt.%. The amount of La and Mg extra addition (8 wt.%) was determined during our basic research (not shown here), to obtain after the MA process, materials with a chemical composition as close as possible to the stoichiometry of an "ideal" reaction. The $La_{1.5}Mg_{0.5}Ni_{7-x}M_x$ (Al; $x = 0$, 0.10, 0.15, 0,20, 0.25 and Mn; $x = 0$, 0.2, 0.3, 0.4, 0.5) powders synthesized by MA were finally heat-treated in 1123 K for 0.5 h in a high purity argon and subsequently cooled in air. For this treatment, the powder (5 g of each composition) was closed under the argon in a quartz tube with a volume of approximately 4 cm$^3$.

The phase analysis crystal structure of synthesized powders was investigated at room temperature by the XRD method (Panalytical, Empyrean model, Almelo, the Netherlands) with CuKα$_1$ ($\lambda = 1.54056$ Å) radiation. The phase quantitative analysis was based on the line profile analysis of the XRD powder patterns realized with the X'Pert High Score Plus software (Tables 2 and 3). The Williamson-Hall (W-H) analysis method was used to study crystallite sizes based on the diffraction pattern of the obtained mechanically alloyed powders.

**Table 1.** Chemical compositions of output powders needed to get the following La-Mg-Ni-M (M = Al, Mn) compounds after all stages of specimen preparation by the application of a SPEX 8000 mixer mill (total weight of milling powders—5 g, the ball to powder mass ratio—4.25:1, milling time—48 h).

| Compound | Chemical Compositions of the Compounds [wt.%] | | | | |
|---|---|---|---|---|---|
| | La | Mg | Ni | Al | Mn |
| $La_{1.5}Mg_{0.5}Ni_7$ | 34.673 | 2.022 | 63.305 | - | - |
| $La_{1.5}Mg_{0.5}Ni_{6.9}Al_{0.1}$ | 34.843 | 2.032 | 62.707 | 0.418 | - |
| $La_{1.5}Mg_{0.5}Ni_{6.85}Al_{0.15}$ | 34.929 | 2.037 | 62.406 | 0.628 | - |
| $La_{1.5}Mg_{0.5}Ni_{6.8}Al_{0.2}$ | 35.015 | 2.042 | 62.103 | 0.840 | - |
| $La_{1.5}Mg_{0.5}Ni_{6.75}Al_{0.25}$ | 35.101 | 2.047 | 61.799 | 1.052 | - |
| $La_{1.5}Mg_{0.5}Ni_{6.8}Mn_{0.2}$ | 34.713 | 2.025 | 61.568 | - | 1.695 |
| $La_{1.5}Mg_{0.5}Ni_{6.7}Mn_{0.3}$ | 34.733 | 2.026 | 60.697 | - | 2.544 |
| $La_{1.5}Mg_{0.5}Ni_{6.6}Mn_{0.4}$ | 34.753 | 2.027 | 59.826 | - | 3.394 |
| $La_{1.5}Mg_{0.5}Ni_{6.5}Mn_{0.5}$ | 34.773 | 2.028 | 58.954 | - | 4.245 |

Pressure-composition isotherms were determined by a Sievert PCI apparatus (Particulate Systems, HPVA 200 model, Norcross, GA, USA). The concentration of the absorbed hydrogen was calculated based on the hydrogen pressure changes measured in the reaction chamber during the tests. The mass of the sample for each measurement cycle was approx. 0.6 g. The investigations of the hydrogen absorption kinetics were carried out at 303 K and under 3 MPa (hydrogen pressure) in the first, second, and third cycle. Each measurement was finished after obtaining the equilibrium hydrogen pressure—the change of pressure did not exceed 200 Pa within 5 min. After each cycle, the samples were degassed at the temperature of 673 K and in a vacuum. The pressure-composition-isotherm (PCI) curves were obtained in the subsequent cycle after the measurements of the kinetics at the same

temperature in the hydrogen pressure range up to approx. 7 MPa. The hydrogen absorption and desorption cycles that occurred during the measurements of the kinetics acted as the activation process. The hydrogen content in the samples was obtained by measuring pressures at constant volumes.

The electrochemical studies were done at room temperature in a three-electrode open cell. The material electrodes in a pellet form ($d$ = 8 mm) consisted of the powder mixture of the synthesized material (0.4 g) and carbonyl nickel (0.04 g). A full description of the electrochemical studies is included in our previous work [7,20]. The electrodes were charged and discharged at a current of 40 mA g$^{-1}$ and the cut-off voltage was −0.7 V vs. the reference Hg/HgO electrode.

## 3. Results

### 3.1. Sample Phase Composition

The series of compounds with the nominal composition $La_{1.5}Mg_{0.5}Ni_{7-x}M_x$ (M = Al; $x$ = 0, 0.10, 0.15, 0,20, 0.25, and M = Mn; $x$ = 0, 0.2, 0.3, 0.4, 0.5) were synthesized by MA (48 h) and subsequently heat-treated at 1123 K for /0.5 h. Figure 1 shows the powder X-ray diffraction patterns and the profile fitting results for $La_{1.5}Mg_{0.5}Ni_{6.85}Al_{0.15}$ and $La_{1.5}Mg_{0.5}Ni_{0.80}Mn_{0.2}$. The synthesized materials are composed of the $La_2Ni_7$ phase crystallizing with the hexagonal and rhombohedral symmetries. The basic $La_{1.5}Mg_{0.5}Ni_7$ composition contains additionally 8 wt.% of LaNi$_5$. In $La_{1.5}Mg_{0.5}Ni_{6.75}Al_{0.25}$, the AB$_3$-type phase (space group: R-3m) is formed. Additionally, traces of $La_2O_3$ in all $La_{1.5}Mg_{0.5}Ni_{7-x}M_x$ (Al and Mn) samples, except $La_{1.5}Mg_{0.5}Ni_{6.85}Al_{0.15}$, are observed (see Tables 2 and 3). In $La_{1.5}Mg_{0.5}Ni_{6.8}Al_{0.2}$, MgO is present. The abovementioned oxide phases like to be formed during the MA process. No pure La, Mg, Ni, Al, or Mn elements are found in the collected XRD patterns. The mean crystallite sizes of produced powders were 37–46 nm according to the XRD analysis.

**Table 2.** Phase abundance in $La_{1.5}Mg_{0.5}Ni_{7-x}Al_x$ [wt.%].

| Compound | $La_{1.5}Mg_{0.5}Ni_7$ | $La_{1.5}Mg_{0.5}$ $Ni_{6.9}Al_{0.1}$ | $La_{1.5}Mg_{0.5}$ $Ni_{6.85}Al_{0.15}$ | $La_{1.5}Mg_{0.5}$ $Ni_{6.8}Al_{0.2}$ | $La_{1.5}Mg_{0.5}$ $Ni_{6.75}Al_{0.25}$ |
|---|---|---|---|---|---|
| $A_2B_7$ P63/mmc | 72.9 | 31.5 | 36.8 | 8.8 | 47.8 |
| $A_2B_7$ R-3m | 16.0 | 63.4 | 63.0 | 77.9 | 42.9 |
| Total of $A_2B_7$ | 88.9 | 94.9 | 99.8 | 86.7 | 90.7 |
| AB$_5$ P6/mmm | 8.0 | - | - | - | |
| AB$_3$ R-3m | - | - | - | - | 1.8 |
| Lanthanum oxide | 3.1 | 4.7 | - | 4.4 | 6.5 |
| Magnesium oxide | | - | - | 0.7 | - |

The published data for 2H-type (P63/mmc) and 3R-type (R3-m) of the $La_{1.5}Mg_{0.5}Ni_7$ phase were used as input data for refinements of our XRD patterns [7]. The analysis results are summarized in Tables 2 and 3. The graphical representation of the line profile analysis of $La_{1.5}Mg_{0.5}Ni_{6.8}Al_{0.2}$ and $La_{1.5}Mg_{0.5}Ni_{6.8}Mn_{0.2}$ are presented in Figure 1. The content of the $A_2B_7$-type phase increased from 88.9% in $La_{1.5}Mg_{0.5}Ni_7$ to 99.8% and 97.8% in $La_{1.5}Mg_{0.5}Ni_{6.8}Al_{0.2}$ and $La_{1.5}Mg_{0.5}Ni_{6.8}Mn_{0.2}$, respectively. However, a reduction in the abundance of the $A_2B_7$-type phase was observed for the Al and Mn concentration above $x$ = 0.15 and 0.3, respectively.

**Table 3.** Phase abundance in $La_{1.5}Mg_{0.5}Ni_{7-x}Mn_x$ [wt.%].

| Compound | $La_{1.5}Mg_{0.5}Ni_7$ | $La_{1.5}Mg_{0.5}$ $Ni_{6.8}Mn_{0.2}$ | $La_{1.5}$ $Mg_{0.5}$ $Ni_{6.7}Mn_{0.3}$ | $La_{1.5}Mg_{0.5}$ $Ni_{6.6}Mn_{0.4}$ | $La_{1.5}Mg_{0.5}$ $Ni_{6.5}Mn_{0.5}$ |
|---|---|---|---|---|---|
| $A_2B_7$ P63/mmc | 72.9 | 40.6 | 41.5 | 39.7 | 42.9 |
| $A_2B_7$ R-3m | 16.0 | 55.3 | 56.3 | 56.9 | 45.1 |
| Total of $A_2B_7$ | 88.9 | 95.9 | 97.8 | 96,6 | 88.0 |
| AB$_5$ P6/mmm | 8.0 | - | - | - | - |
| Lanthanum oxide | 3.1 | 4.1 | 2.2 | 3.4 | 12 |

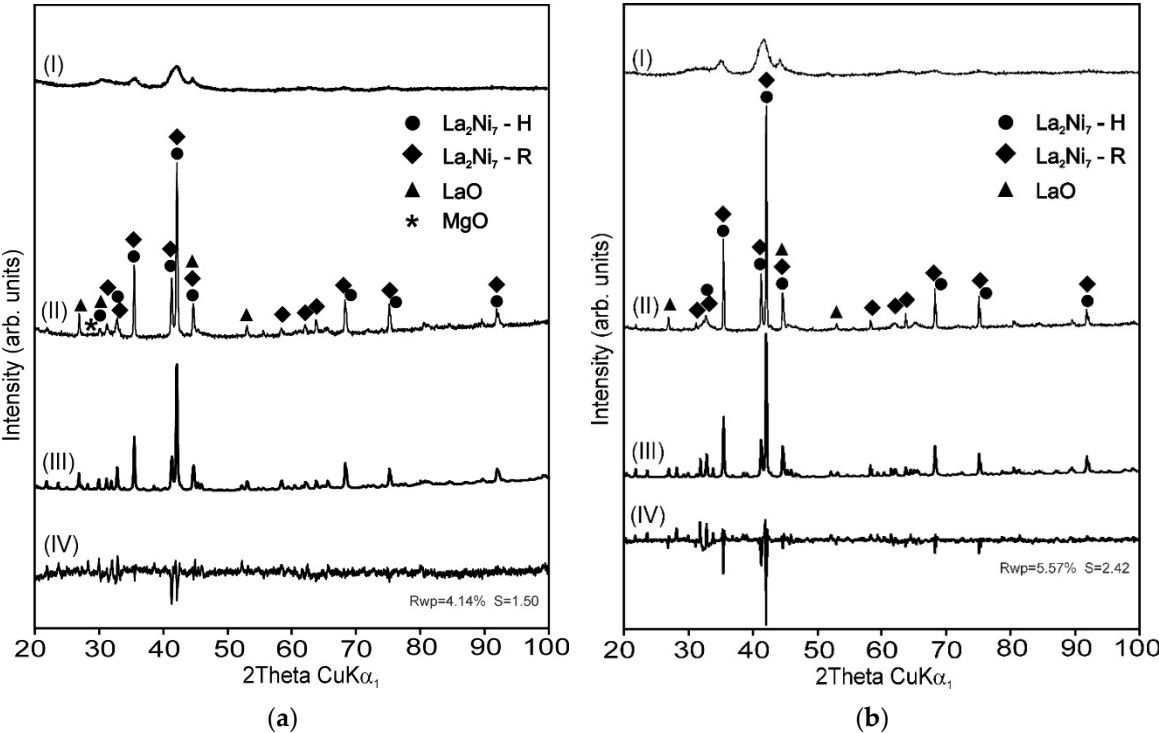

**Figure 1.** X-ray diffraction patterns of: (**a**) $La_{1.5}Mg_{0.5}Ni_{6.8}Al_{0.2}$ and (**b**) $La_{1.5}Mg_{0.5}Ni_{6.80}Mn_{0.2}$. After MA (I), after annealing at 1123 K/0.5 h (II), profile fitting-calculated pattern (III), and differences between calculated and observed patterns (IV).

Recently, the influence of the Ni replacement with Mn on the phase composition of the $ReNi_{2.6−x}Mn_xCo_{0.9}$ ($x$ = 0.0, 0.225, 0.45, 0.675, 0.90) alloys synthesized by induction melting were studied [27]. In these alloys, $(La,Ce)_2Ni_7$ phase, $(Pr,Ce)Co_3$ phase, and $(La,Pr)Ni_5$ phase were the main phases. Independently, the effects of the partial Ni replacement by Al on the microstructures of the as-cast and rapidly quenched $La_{0.7}Mg_{0.3}Ni_{2.55−x}Co_{0.45}Al_x$ ($x$ = 0, 0.1, 0.2, 0.3, 0.4) were studied [28]. Both, the as-cast and quenched samples consist of multiple phases including $(La,Mg)Ni_3$, $LaNi_5$, and $LaNi_2$. In general, the synthesis method of the $La_{1.5}Mg_{0.5}Ni_7$-type hydrogen storage alloys influences strongly their final phase composition.

### 3.2. Thermodynamic Properties

The hydrogen absorption-desorption behavior of $La_{1.5}Mg_{0.5}Ni_{7−x}M_x$ (M = Al ($0 \leq x \leq 0.25$), Mn ($0 \leq x \leq 0.5$)) are outlined in Tables 4 and 5. Figure 2 shows a correlation between the PCI data measured for the synthesized materials.

All alloys absorb hydrogen at 303 K. The shift from the α-solid solution to the β-hydride phase is observed. The absorption plateau pressure for the $La_{1.5}Mg_{0.5}Ni_{7−x}M_x$ (M = Al and Mn) is much lower than that for $La_{1.5}Mg_{0.5}Ni_7$. The Al or Mn contents in the studied system influenced the hydrogen sorption pressure. Due to the higher stability of the $La_{1.5}Mg_{0.5}Ni_{7−x}M_x$ (M = Al and Mn) hydrides, a decrease of the sorption pressure was measured for higher Al or Mn contents in the $La_{1.5}Mg_{0.5}Ni_{7−x}$ system. Previously, it was shown that the plateau pressure during the sorption increases with the increasing Mg content in $La_{2−x}Mg_xNi_7$ [7]. Additionally, the substitution of La by Mg results in the stability decrease of $(La,Mg)_2Ni_7$ hydrides [35]. It is important to note, that the hydrogen sorption plateau pressure close to the atmospheric one was obtained in $La_{2−x}Mg_xNi_7$ for $x$ = 0.25 and 0.5 [7].

Independently, Mani et al. observed that plateau pressure on the hydrogen sorption in the La-Mg-Ni-based systems decreased with the incorporation not only of Al, Mn but also of V, Cu, Fe, and Co [36]. The replacement of nickel by Mn, Al, and Co in $La_{1.5}Mg_{0.5}Ni_{7−x}M_x$ has been studied to

find the optimum concentration of the substituting elements that would ensure the corrosion resistance and high hydrogen capacity.

**Table 4.** Thermodynamic properties of $La_{1.5}Mg_{0.5}Ni_{7-x}Al_x$.

| Compound | $La_{1.5}Mg_{0.5}$ $Ni_{7.0}$ | $La_{1.5}Mg_{0.5}$ $Ni_{6.9}Al_{0.1}$ | $La_{1.5}Mg_{0.5}$ $Ni_{6.85}Al_{0.15}$ | $La_{1.5}Mg_{0.5}$ $Ni_{6.8}Al_{0.2}$ | $La_{1.5}Mg_{0.5}$ $Ni_{6.75}Al_{0.25}$ |
|---|---|---|---|---|---|
| Maximum hydrogen capacity [wt.%] | 1.53 | 1.58 | 1.34 | 1.37 | 1.57 |
| Equilibrium pressure of hydrogen absorption [MPa] | 0.14 | 0.064 | 0.040 | 0.014 | 0.018 |
| The time required to achieve 95% of maximum hydrogen capacity on the third cycles [min] | 13 | 5 | 10 | 7 | 7 |

**Table 5.** Thermodynamic properties of $La_{1.5}Mg_{0.5}Ni_{7-x}Mn_x$.

| Compound | $La_{1.5}Mg_{0.5}$ $Ni_7$ | $La_{1.5}Mg_{0.5}$ $Ni_{6.8}Mn_{0.2}$ | $La_{1.5}Mg_{0.5}$ $Ni_{6.7}Mn_{0.3}$ | $La_{1.5}Mg_{0.5}$ $Ni_{6.6}Mn_{0.4}$ | $La_{1.5}Mg_{0.5}$ $Ni_{6.5}Mn_{0.5}$ |
|---|---|---|---|---|---|
| Maximum hydrogen capacity [wt.%] | 1.53 | 1.79 | 1.76 | 1.58 | 1.30 |
| Equilibrium pressure of hydrogen absorption [MPa] | 0.14 | 0.025 | 0.018 | 0.017 | 0.015 |
| The time required to achieve 95% of maximum hydrogen capacity on the 3rd cycle [min] | 13 | 6 | 10 | 14 | 39 |

Table 6 shows the time-capacity kinetics curves for hydrogen absorption for $La_{1.5}Mg_{0.5}Ni_7$ and $La_{1.5}Mg_{0.5}Ni_{7-x}M_x$. The data were obtained at 303 K for the first three cycles. To reach the maximum hydrogen storage capacity and best kinetic properties the activation process of the samples was performed (see Table 6). It is important to note, that the chemical modification of Ni by Al or Mn in this study affected the kinetics of the hydrogen sorption. The $La_{1.5}Mg_{0.5}Ni_7$ absorbs 95% of the maximum hydrogen volume in 301 min, while the best Al-containing sample needs only 4 min ($x = 0.1$). After chemical modification of the alloy by Al or Mn substitution, the maximum hydrogen storage capacity was increased (Figure 2). The highest value of 1.79 wt.% was obtained for the $La_{1.5}Mg_{0.5}Ni_{6,80}Mn_{0.20}$.

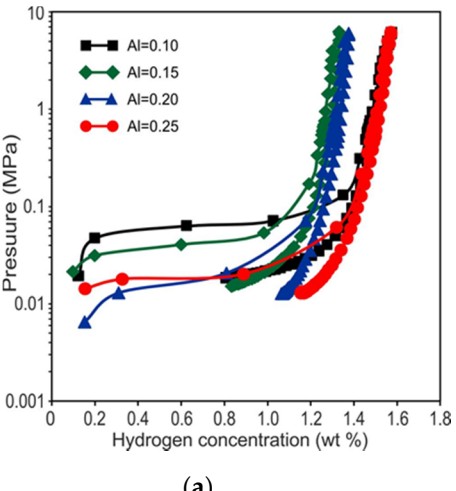

(**a**)

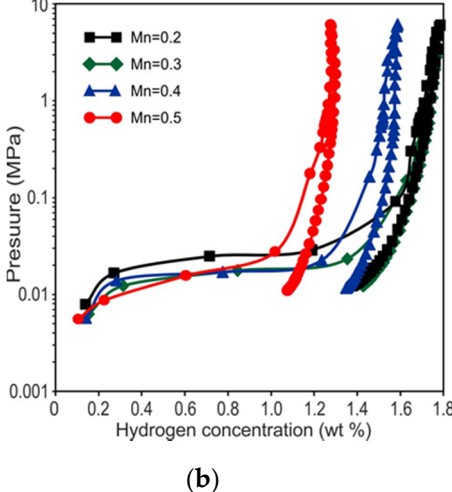

(**b**)

**Figure 2.** Sorption curves measured at 303 K for (**a**) $La_{1.5}Mg_{0.5}Ni_{7-x}Al_x$ and (**b**) $La_{1.5}Mg_{0.5}Mn_xNi_{7-x}$.

**Table 6.** Hydrogen concentration from kinetic measurements for $La_{1.5}Mg_{0.5}Ni_{7-x}Al_x$ and $La_{1.5}Mg_{0.5}Ni_{7-x}Mn_x$ (measurement uncertainty ±1.5%).

| Compound | Kinetic Measurements | | | | | |
|---|---|---|---|---|---|---|
| | Hydrogen Concentration [wt.%] | | | The Time Needed to Reach 95% of the Maximum Hydrogen Capacity [min] | | |
| | 1 Cycle | 2 Cycle | 3 Cycle | 1 Cycle | 2 Cycle | 3 Cycle |
| $La_{1.5}Mg_{0.5}Ni_7$ | 1.60 | 1.59 | 1.59 | 301 | 13 | 13 |
| $La_{1.5}Mg_{0.5}Ni_{6.9}Al_{0.1}$ | 1.51 | 1.52 | 1.52 | 4 | 6 | 5 |
| $La_{1.5}Mg_{0.5}Ni_{6.85}Al_{0.15}$ | 0.88 | 1.43 | 1.41 | 134 | 18 | 10 |
| $La_{1.5}Mg_{0.5}Ni_{6.8}Al_{0.2}$ | 1.53 | 1.45 | 1.39 | 12 | 6 | 7 |
| $La_{1.5}Mg_{0.5}Ni_{6.75}Al_{0.25}$ | 1.54 | 1.53 | 1.54 | 10 | 9 | 7 |
| $La_{1.5}Mg_{0.5}Ni_{6.8}Mn_{0.2}$ | 1.71 | 1.67 | 1.73 | 183 | 8 | 6 |
| $La_{1.5}Mg_{0.5}Ni_{6.7}Mn_{0.3}$ | 1.07 | 1.75 | 1.73 | 78 | 40 | 10 |
| $La_{1.5}Mg_{0.5}Ni_{6.6}Mn_{0.4}$ | 0.95 | 1.56 | 1.58 | 142 | 17 | 14 |
| $La_{1.5}Mg_{0.5}Ni_{6.5}Mn_{0.5}$ | 0.72 | 1.34 | 1.32 | 140 | 31 | 39 |

*3.3. Electrochemical Properties*

The electrochemical properties of the studied $La_{1.5}Mg_{0.5}Ni_{7-x}M_x$ (Al; $x$ = 0, 0.10, 0.15, 0,20, 0.25) and Mn; $x$ = 0, 0.2, 0.3, 0.4, 0.5) have been summarized in Figures 3 and 4. The most distinguished data have been presented in Tables 7 and 8. It is important to note, that the electrodes show the maximum capacities in the 3rd cycle. For the $La_{1.5}Mg_{0.5}Ni_{6.85}Al_{0.15}$ electrode, the highest obtained discharge capacity was 328 mAh/g.

**Table 7.** The electrochemical properties of $La_{1.5}Mg_{0.5}Ni_{7-x}Al_x$.

| Compound | $La_{1.5}Mg_{0.5}$ $Ni_7$ | $La_{1.5}Mg_{0.5}$ $Ni_{6.9}Al_{0.1}$ | $La_{1.5}Mg_{0.5}$ $Ni_{6.85}Al_{0.15}$ | $La_{1.5}Mg_{0.5}$ $Ni_{6.8}Al_{0.2}$ | $La_{1.5}Mg_{0.5}$ $Ni_{6.75}Al_{0.25}$ |
|---|---|---|---|---|---|
| Maximum discharge capacity [mAh/g] | 304 | 304 | 328 | 277 | 243 |
| Discharge capacity at 30th cycle [mAh/g] | 204 | 220 | 234 | 211 | 183 |
| Discharge capacity at 50th cycle [mAh/g] | 167 | 189 | 203 | 184 | 144 |
| $C_{30}/C_{max} \times 100\%$ * | 67.1 | 72.6 | 71.4 | 76.3 | 75.4 |
| $C_{50}/C_{max} \times 100\%$ * | 54.9 | 62.2 | 61.9 | 66.3 | 59.2 |

* $C_{30}$ and $C_{50}$ are the discharge capacities after the 30th and 50th charge/discharge cycles, $C_{max}$ is the maximum discharge capacity.

The discharge capacities of all the studied $La_{1.5}Mg_{0.5}Ni_{7-x}M_x$ electrode materials degraded during the charge-discharge cycling, most likely to partial oxidation of the electrode materials or the formation of the stable hydride phases. The origin of this behavior could be the formation of the $Mg(OH)_2$ and $La(OH)_3$ layers on the surface. These layers decrease the surface electrocatalytic activity and prevent hydrogen diffusion into the electrodes. The pulverization of the electrodes during the hydrogenation and dehydrogenation cycles also influence the electrochemical properties.

The cycle stability of the $La_{1.5}Mg_{0.5}Ni_{7-x}Al_x$ (Al ($x$ = 0.10, 0.15, 0.20) and Mn ($x$ = 0.2, 0.3) electrodes increases. Up to now, the best cycle stability was observed in $La_{2-x}Mg_xNi_7$ [7]. The phase composition of this material influenced strongly its electrochemical properties. The major phase in the $LaMgNi_7$ sample is $LaNi_5$ [7]. This material has a higher electrochemical cycle stability in comparison with $(La, Mg)_2Ni_7$ [37]. On the other hand, $La_{1.5}Mg_{0.5}Ni_{7-x}Al_{0.2}$ has the best capacity retaining rate after the 50th cycle. It is important to note, that the partial substitution of nickel with aluminum or manganese resulted in the increased cycle stability of the $MH_x$ alloy electrodes. Additionally,

these chemical modifications also influenced the kinetics of the hydrogen sorption reducing the time of the hydrogenation process.

**Table 8.** The electrochemical properties of $La_{1.5}Mg_{0.5}Ni_{7-x}Mn_x$.

| Compound | $La_{1.5}Mg_{0.5}Ni_7$ | $La_{1.5}Mg_{0.5}Ni_{6.8}Mn_{0.2}$ | $La_{1.5}Mg_{0.5}Ni_{6.7}Mn_{0.3}$ | $La_{1.5}Mg_{0.5}Ni_{6.6}Mn_{0.4}$ | $La_{1.5}Mg_{0.5}Ni_{6.5}Mn_{0.5}$ |
|---|---|---|---|---|---|
| Maximum discharge capacity [mAh/g] | 304 | 284 | 300 | 282 | 271 |
| Discharge capacity at 30th cycle [mAh/g] | 204 | 209 | 225 | 187 | 179 |
| Discharge capacity at 50th cycle [mAh/g] | 167 | 185 | 197 | 154 | 146 |
| $C_{30}/C_{max} \times 100\%$ * | 67.1 | 73.5 | 74.9 | 66.5 | 66.0 |
| $C_{50}/C_{max} \times 100\%$ * | 54.9 | 65.0 | 65.7 | 54.5 | 53.8 |

\* $C_{30}$ and $C_{50}$ are the discharge capacities after 30th and 50th charge/discharge cycles, $C_{max}$ is the maximum discharge capacity.

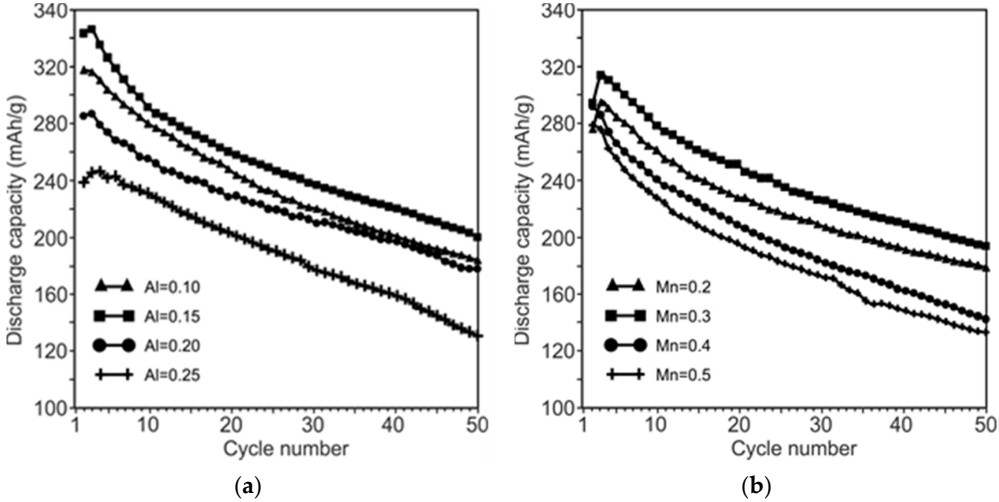

**Figure 3.** Discharge capacities as a function of cycle number of (**a**) $La_{1.5}Mg_{0.5}Ni_{7-x}Al_x$ and (**b**) $La_{1.5}Mg_{0.5}Ni_{7-x}Mn_x$.

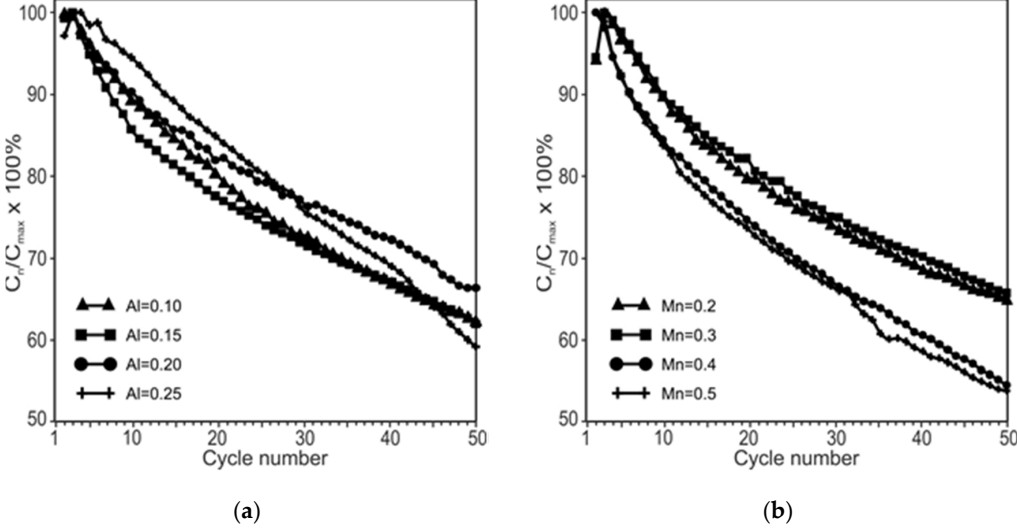

**Figure 4.** Cyclic stability of (**a**) $La_{1.5}Mg_{0.5}Ni_{7-x}Al_x$ and (**b**) $La_{1.5}Mg_{0.5}Ni_{7-x}Mn_x$ alloys.

## 4. Discussion

Recently, the research was directed to the new generation of hydrogen storage (La, Mg)$_2$Ni$_7$ materials [2,15,19,24]. These hydrogen storage phases could replace the poor cycle stability of the ZrV$_2$- and LaNi$_5$-type hydrides [15,38]. Many different ways of synthesis of nanostructured hydrogen storage materials are available [39]. The mechanical processes include mechanical alloying or high energy ball milling. MA is an effective process to produce the (La-Mg)$_2$Ni$_7$ alloys with reduced crystallite sizes and fresh surfaces. MA can improve the kinetics of hydrogen absorption and desorption of the processed materials due to large surface areas and as a consequence short hydrogen diffusion pathways. For example, the TiV alloy synthesized by MA shows a multi-crystalline microstructure [34].

The main purpose of our current study is the synthesis of new (La,Mg)$_2$Ni$_7$-type hydrogen storage alloys via its chemical modification. The effect of the different metals on the phase compositions as well as thermodynamic and electrochemical properties of this system was studied. In the La-Mg-Ni-type system, various crystalline phases could be formed, among which (La,Mg)Ni$_3$, (La,Mg)$_2$Ni$_7$, and (La,Mg)$_5$Ni$_{19}$ are observed [40,41]. They are composed of the [A$_2$B$_4$] and [AB$_5$] subunits alternatively stacking along the *c* axis [15]. Studies on the thermodynamic and electrochemical behavior of the La$_2$Ni$_7$-type compounds show that the additional presence of (La,Mg)$_5$Ni$_{19}$ or LaNi$_5$ phase has a positive catalytic effect on the charge-discharge process of this alloy [42]. The transitional metals affected the hydrogen absorption/desorption plateau pressure of hydrogen storage materials and influenced their thermodynamic and electrochemical properties [37,43–45].

In this work, the influence of the Al and Mn concentration in the A$_2$B$_7$-type (La$_{1.5}$Mg$_{0.5}$Ni$_{7-x}$M$_x$ (M = Al ($0 \leq x \leq 0.25$), Mn ($0 \leq x \leq 0.5$)) materials, synthesized by MA, on the thermodynamic and electrochemical properties was studied. All of the hydrogen storage materials are composed of the La$_2$Ni$_7$ phase (the hexagonal structure—Ce$_2$Ni$_7$-type and the rhombohedral structure—Gd$_2$Co$_7$-type). In La$_{1.5}$Mg$_{0.5}$Ni$_{6.75}$Al$_{0.25}$, the AB$_3$-type phase was detected, too. Additionally, traces of the La$_2$O$_3$ phase in some of the La$_{1.5}$Mg$_{0.5}$Ni$_{7-x}$M$_x$ (Al and Mn) alloys were observed. In the La$_{1.5}$Mg$_{0.5}$Ni$_{6.8}$Al$_{0.2}$ alloy, the MgO phase was viewed. The abundance of the La$_2$Ni$_7$-type phase increased from 88.9% (La$_{1.5}$Mg$_{0.5}$Ni$_7$ alloy) to 99.8% and 97.8% for La$_{1.5}$Mg$_{0.5}$Ni$_{6.85}$Al$_{0.15}$ and La$_{1.5}$Mg$_{0.5}$Ni$_{6.7}$Mn$_{0.3}$, respectively. The maximum value of the La$_2$Ni$_7$ phase is observed in the La$_{1.5}$Mg$_{0.5}$Mg$_{0.5}$Ni$_{6.85}$Al$_{0.15}$ alloy (Table 2).

All studied alloys absorb hydrogen at 303 K. The shift from the α-solid solution to the β-hydride phase is observed. The absorption plateau pressure for the La$_{1.5}$Mg$_{0.5}$Ni$_{7-x}$M$_x$ (M = Al and Mn) are much lower than for La$_{1.5}$Mg$_{0.5}$Ni$_7$. The hydrogen sorption pressure of the studied system depends on the Al or Mn contents. Due to the higher stability of the La$_{1.5}$Mg$_{0.5}$Ni$_{7-x}$M$_x$ (M = Al and Mn) hydrides, a decrease of the sorption pressure was observed for higher Al or Mn contents in the La$_{1.5}$Mg$_{0.5}$Ni$_{7-x}$M$_x$ system. The highest value of hydrogen content and the discharge capacity was measured for La$_{1.5}$Mg$_{0.5}$Ni$_{6.8}$Mn$_{0.2}$ (1.79 wt.%) and La$_{1.5}$Mg$_{0.5}$Ni$_{6.85}$Al$_{0.15}$ (328 mAh/g), respectively. On the other hand, the substitution of Ni with Al or Mn in the MA and annealed La$_{1.5}$Mg$_{0.5}$Ni$_{7-x}$M$_x$ (Al; *x* = 0.10, 0.15, 0,20, 0.25, and Mn; *x* = 0.2, 0.3) improved the cycle stability of the synthesized electrodes. Additionally, the stability of the electrochemical discharge capacity increases with the increasing content of Al and Mn up to *x* = 0.2 and 0.3, respectively. However, a significant reduction in the discharge capacity was measured for the Al and Mn content above *x* = 0.25 and 0.5, respectively.

An additional increase of hydrogenation properties of these hydrogen storage materials can be established by encapsulation of alloy particles with thin amorphous nickel coating [19]. Modification of the La$_{1.5}$Mg$_{0.5}$Ni$_7$ particles with an electroless, 1 m thick Ni-P coating weaken the electrodes corrosion process.

## 5. Conclusions

In this work, a series of La$_{1.5}$Mg$_{0.5}$Ni$_{7-x}$M$_x$ (M = Al ($0 \leq x \leq 0.25$), Mn ($0 \leq x \leq 0.5$)) intermetallics were prepared using mechanical alloying and heat treatment. The effect of the substitution of Ni with Al and Mn on the sample phase compositions, thermodynamic and electrochemical properties of the

$A_2B_7$-type La-Mg-Ni-M-based materials was investigated. Partial replacement of Ni with Al and Mn in the La-Mg-Ni alloy improved the hydrogen sorption of this system. For the $La_{1.5}Mg_{0.5}Ni_{6.9}Al_{0.1}$ and $La_{1.5}Mg_{0.5}Ni_{6.8}Mn_{0.2}$ alloys, the time required to get 95% of the maximum hydrogen capacity at the 3rd cycle decreases to reach 5 and 6 min, respectively. On the other hand, the $La_{1.5}Mg_{0.5}Ni_{6.85}Al_{0.15}$ alloy has a high discharge capacity, then that of the $La_{1.5}Mg_{0.5}Ni_7$. Moreover, the change of Ni with Al or Mn in $La_{1.5}Mg_{0.5}Ni_{7-x}M_x$ also enhanced the stability of the discharge capacity. Generally, the improvement of the properties of the La-Mg-Ni-M-based hydrogen storage alloys discussed in this manuscript is the function of the phase composition and final microstructure of the synthesized hydrogen storage material.

**Author Contributions:** M.N., M.B., and M.J. conducted the experimental and analytical works as well as wrote the manuscript; M.J. supervised the project. All authors contributed to the critical reading and editing of the final version of the manuscript.

**Funding:** The work has been financed by the National Science Centre, Poland under the decision no. 014/15/B/ST8/00088.

**Conflicts of Interest:** The authors declare no conflict of interest.

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
