# Peer review of "Effect of Substitutional Elements on the Thermodynamic and Electrochemical Properties of Mechanically Alloyed La1.5Mg0.5Ni7−xMx alloys (M = Al, Mn)"

_metals, doi:10.3390/met10050578_

Round 1

Reviewer 1 Report

The manuscript focuses on a hot subject and represents a good experimental work.

However it could be improved in quality of presentation and introduction background.

Author Response

Dear Reviewer 1,
We would like to appreciate you giving us the opportunity to resubmit our manuscript: ” Effect of substitutional elements on the thermodynamic and electrochemical properties of mechanically alloyed La1.5Mg0.5Ni7-xMx alloys (M= Al, Mn)” by M. Nowak, M. Balcerzak, M. Jurczyk to Metals. We are pleased with your comments and objective feedback considering the draft of our article. The changes were marked in our article in red colors.
All of the suggestions were included during the correction of our article and the comments were appropriately responded:
Review comment:
The manuscript focuses on a hot subject and represents a good experimental work. However, it could be improved in the quality of presentation and introduction background.
Response:
We are deeply grateful to the suggestions by the reviewer which has enabled us to improve our manuscript. We have revised our manuscript conscientiously. Changes within the manuscript are marked in red.
We would like to thank you once again for your suggestions for improving our manuscript.

Yours faithfully,
M. Jurczyk

Reviewer 2 Report

The manuscript entitled "Effect of substitutional elements on the thermodynamic and electrochemical properties of mechanically alloyed La1.5Mg0.5Ni7-xMx alloys (M= Al, Mn)" reports on thermodynamic, kinetic and electrochemical properties of the A2B7-typ phases, obtained by mechanical alloying, in the La-Mg-Ni-Al/La-Mg-Ni-Mn systems. Though, the study appears quite interesting the results are poorly presented. The structure of the paper is very chaotic with randomly linked results. All sections require significant modifications. Apart from that, the language should be improved. The detailed comments and suggestions are included in the attached PDF file.

Author Response

Dear Reviewer 2,

We are deeply grateful to the suggestions by the reviewer which has enabled us to improve our manuscript. Based on their comments, we have revised our manuscript conscientiously. Changes within the manuscript are marked in red and our point-by-point responses to each comment are listed below.

Extensive editing of English language and style required

Response - The language was improved.

The manuscript entitled "Effect of substitutional elements on the thermodynamic and electrochemical properties of mechanically alloyed La1.5Mg0.5Ni7-xMx alloys (M= Al, Mn)" reports on thermodynamic, kinetic and electrochemical properties of the A2B7-typ phases, obtained by mechanical alloying, in the La-Mg-Ni-Al/La-Mg-Ni-Mn systems. Though, the study appears quite interesting the results are poorly presented. The structure of the paper is very chaotic with randomly linked results. All sections require significant modifications. Apart from that, the language should be improved. The detailed comments and suggestions are included in the attached PDF file.

Changes within the manuscript are marked in red and our point-by-point responses to each comment are listed below.

Line 106

Please add detailed information about milling conditions, i.e. milling media materials, vial volume, ball-to-powder weight ratio, milling frequency, ...

Response – lines 97-100:   Mechanical alloying was carried out using a 8000 SPEX mixer mill (SPEX SamplePrep, Metuchen, USA) with milling frequency 875 Hz, employing a weight ratio of hard steel balls to powder weight ratio of 4.25:1 at ambient temperature for 48 h in a continuous mode.

Line 111

Please add details regarding the thermal treatment, i.e. what was the form of the annealed samples (compacted or loose?), were powders sealed in a tube? if so, what kind of tube? what was the pressure during annealing? could that effect the Mg and Mn evaporation? What was the atmosphere?

Response – lines 103-110:   The elemental powders were weighed, blended, and poured into a round bottom stainless vials (35 mL) in a glove box (Labmaster 130) filled with automatically controlled argon atmosphere (O2 ≤ 2 ppm and H2O ≤ 1 ppm) to obtain the materials. A composition of starting materials mixture was based on the stoichiometry of an “ideal” reaction. However, due to oxidation of La and Mg, the content of theses element was increased by 8 wt%. The amount of La and Mg extra addition (8 wt. %) was determined during our basic research (not shown here), to obtain after the MA process, materials with chemical composition as close as it is possible to the stoichiometry of an “ideal” reaction.

Line 114

If it was Cu K alpha then the wavelength is wrong, since 1.54056 Å refers to Cu K alpha1. Please clarify also in the figures. This is also important in view of the structural analysis the authors performed. A2B7 compunds should be studied by a monochromatic X-ray radiation due to the similar symmetry and unit cell parameters of the coexisting, typically formed other superlattice compunds.

Response – lines 114 – 115: The crystal structure of synthesized powders was investigated at room temperature by the XRD method (Panalytical, Empyrean model, Almelo, Netherlands) with CuKα1 (l = 1.54056 Å) radiation.  

Line 116

Could the authors include some results?

Response: The scanning electron microscopy and EDS data are not included in this study – see our previous work:

[7] Balcerzak, M.; Nowak, M.; Jurczyk, M. Hydrogenation and electrochemical studies of La-Mg-Ni alloys. Int. J. Hydrogen Energy 2017, 42, 1436-1443, doi.org/10.1016/j.ijhydene.2016.05.220.

Line 145

Does it mean that the authors did not include Al/Mn atoms during the refinements? How do we know that the substitution took place? It should be clearly stated in the text. Please include values of the refined atomic postilions + esd. Also, the extended description of the refinement process should be given. It is important to report the values of the refined unit cell parameters for the formed phases and compare the calculated values with those available in the literature, including the parent (La,Mg)2Ni7. Especially with compositions fabricated by different synthesis method. The variation in the volume of A2B7 is expected if the Al/Mn substitution indeed took place. What should be expected from the analysis of the differences between Ni, Al and Mn atomic radii? How these predictions correspond with the calculated values of the unit cell volumes? The discrepancies between the presented in Fig. 1 observed and calculated PXD patterns are huge. They clearly indicate that either the refined crystal structure model is incorrect (not complete) or, what is more likely, there are other superlattice compunds, with similar symmetries, present in the samples (AB3, A5B19, etc), which cannot be distinguish due to the peak overlap among them. So, any statements related to the sample phase competitions and the phase structural properties must be said very carefully!

Response:   We agree with this comment. The mass fractions of the phases in the La1.5Mg0.5Ni7-xMx compounds were obtained from the Rietveld refinements

Lines 114 – 119: The crystal structure of synthesized powders was investigated at room temperature by the XRD method (Panalytical, Empyrean model, Almelo, Netherlands) with CuKα1 (l = 1.54056 Å) radiation. The phase quantitative analysis was based on the Rietveld profile fitting method realized on the X'Pert High Score Plus software. (Tables 2 and 3). The Williamson-Hall (W-H) analysis method was used to study crystallite size on the peak broadening of the obtained mechanical alloyed powders spectra.

In new Table 1 the chemical compositions of output powders needed to get following La-Mg-Ni-M (M=Al, Mn) compounds after all stages of specimen preparation by the application of SPEX 8000 Mixer Mill (total weight of milling powders – 5 g, the ball to powder mass ratio – 4.25:1, milling time – 48 h) is given.

Line 246

Why this is not reported in the result section? How the analysis was carried out? The numbers refer to which series?

Response:   lines 117-118 – The Williamson-Hall (W-H) analysis method was used to study crystallite size on the peak broadening of the obtained mechanical alloyed powders spectra.

and lines 152-153 - The mean crystallite sizes of produced powders were 37 - 46 nm according to XRD analysis.

And finally, new references were added – see [8-14]

  • Denys, R.V.; Riabov A.B.; Yartys V.A.; Masashi Sato; Delaplane R.G. Mg substitution effect on the hydrogenation behaviour, thermodynamic and structural properties of the La2Ni7–H(D)2 system, J. Solid State Chem. 2008, 181, 812-821, doi.org/10.1016/j.jssc.2007.12.041.
  • Guzik, M.N.; Hauback, B.C.; Yvon, K. Hydrogen atom distribution and hydrogen induced site depopulation for the La2−xMgxNi7–H system, J. Solid State Chem. 2012, 186, Pages 9-16, doi.org/10.1016/j.jssc.2011.11.026.
  • Ozaki, T.; Kanemoto, M.; Kakeya, T.; Kitano, Y.; Kuzuhara, M.; Watada, M.; Tanase, T.Sakai, T. Stacking structures and electrode performances of rare earth–Mg–Ni-based alloys for advanced nickel–metal hydride battery. J. Alloys Compd. 2007, 446–447, 620-624, doi.org/10.1016/j.jallcom.2007.03.059.
  • Guzik, M.N.; Lang, J.; Huot, J.; Sartori, S. Effect of Al presence and synthesis method on phase composition of the hydrogen absorbing La–Mg–Ni-based compounds. Int. J. Hydrogen Energy 2017, 42, 30135-30144, doi.org/10.1016/j.ijhydene.2017.10.062.
  • Férey, A.; Cuevas, F.; Latroche, M.; Knosp, B.; Bernard, P. Elaboration and characterization of magnesium-substituted La5Ni19 hydride forming alloys as active materials for negative electrode in Ni-MH battery. Electrochim. Acta 2009, 54, 15, 1710-1714, doi.org/10.1016/j.electacta.2008.09.069.
  • Nakamura, J.; Iwase, K.; Hayakawa, H.; Nakamura, Y.; Akib, E. Structural Study of La4MgNi19 Hydride by In Situ X-ray and Neutron Powder Diffraction, J. Phys. Chem. C 2009, 113, 5853-5859, doi.org/10.1021/jp809890e.
  • Filinchuk, Y.E.; Yvon, K.; Emerich, H. Tetrahedral D Atom Coordination of Nickel and Evidence for Anti-isostructural Phase Transition in Orthorhombic Ce2Ni7D4, Inorg. Chem. 2007, 46, 2914-2920, doi.org/10.1021/ic062312u

We would like to thank you once again for your suggestions for improving our manuscript.

Yours faithfully,

  1. Jurczyk

Reviewer 3 Report

The paper by Nowak et al. reports an extensive study of the hydrogenation and electrochemical properties of novel La1.5Mg0.5Ni7-xMx alloys (M= Al,  Mn) synthesized by mechanical alloying and subsequent thermal treatments.
The paper is interesting and reports original investigations and, therefore, it deserves publication. However, I recommend some major changes before eventual publication:
- please, revise English in the whole manuscript;
- revise the sentence in lines 35-36, that I do not understand ("is prime to");
- define C100 in line 56;
- Lines 70-73: do the author talk about hydrogen storage or electrochemical properties?
- revise lines 74-75;
- Line 79: melt spinning -> melt spun
- Lines 110-111: powders synthesized by MA for 48 h were finally heat-treated in 1123 K for 0.5h in high purity argon: was the MA process continuous or were there periodic stops? What was the temperature rate to reach 1123 K and to come back to room temperature?
- Line 143 & 151: contentment -> content
- Lines 145-146: "the Rietveld refinements of La1.5Mg0.5Ni6.85Al0.15 and La1.5Mg0.5Ni6.7Mn0.3 alloys are presented in Fig 1", from the caption of the figure it seems that the Al and Mn content are 0.2 in both alloys.
- Lines 164 and subsequent: did the authors perform any activation procedure for the alloys in order to absorb hydrogen? Which one? Were absorption/desorption cycles performed more than three times (even without measuring the p-c curves but just the hydrogen capacity)?
- The absorption/desorption curves are not perfectly horizontal. How was the plateau pressure defined? Did the authors consider the middle point of the absorption curves?
- Figure 2: the use of colors to differentiate the curves would help the readers; moreover, the absorption curves extend up to about 70-80 MPa that is 700-800 bar. Are the authors sure? The apparatus they used for those measurements cannot attain these pressure values.
- Table 6 & 7: Define C30 and C50;
- Lines 212-213: I do not understand the sentence:"The cycle stability of the La 1.5 Mg 0.5 Ni 7 x M x Al x = 0.10, 0.15, 0.20 and Mn x = 0.2, 0.3 ) electrodes increases." How was stability defined?
- Lines 268-272: Is this investigation part of the present work or was it a previous investigation (please, cite).

Author Response

Dear Reviewer 3,

We would like to appreciate you giving us the opportunity to resubmit our manuscript: ” Effect of substitutional elements on the thermodynamic and electrochemical properties of mechanically alloyed La1.5Mg0.5Ni7-xMx alloys (M= Al, Mn)” by M. Nowak, M. Balcerzak, M. Jurczyk to Metals. We are pleased with your comments and objective feedback considering the draft of our article. The changes were marked in our article in red colors.

All of the suggestions were included during the correction of our article and the comments were appropriately responded:

Moderate English changes required. The paper by Nowak et al. reports an extensive study of the hydrogenation and electrochemical properties of novel La1.5Mg0.5Ni7-xMx alloys (M= Al,  Mn) synthesized by mechanical alloying and subsequent thermal treatments.
The paper is interesting and reports original investigations and, therefore, it deserves publication.

However, I recommend some major changes before eventual publication:

- please, revise English in the whole manuscript;
Response - The language was improved.

- revise the sentence in lines 35-36, that I do not understand ("is prime to");
Response – Generally, the hydrogen storage properties of ternary La-Mg-Ni hydrides are superior to corresponding binary ABn (n = 2-5) [1, 5, 6, 7, 15-21].

- define C100 in line 56;
Response – lines 56-57: (C100 is the discharge capacity after 100th charge/discharge cycles, Cmax is the maximum discharge capacity)

- Lines 70-73: do the author talk about hydrogen storage or electrochemical properties?
Response – Yes, only electrochemical properties

- revise lines 74-75;

Response – The substitution of Ni with Co, Mn, Fe, Al, and Cu in La 2MgNi9 decreases the hydrogen storage capacity, but at the same time increases the hydride stability [29].

- Line 79: melt spinning -> melt spun

Response – lines 78-80: The effects of the partial Ni replacement by Fe, Mn, and Al on the microstructures and electrochemical properties of La0.7Mg0.3Ni2.55-xCo0.45Mx (M = Fe, Mn, Al; x = 0, 0.1), synthesized by melt spinning, was studied by Zhang et al. [30].

- Lines 110-111: powders synthesized by MA for 48 h were finally heat-treated in 1123 K for 0.5h in high purity argon: was the MA process continuous or were there periodic stops? What was the temperature rate to reach 1123 K and to come back to room temperature?

Response – lines 97-100: Mechanical alloying was carried out using a 8000 SPEX mixer mill (SPEX SamplePrep, Metuchen, USA) with milling frequency 875 Hz, employing a weight ratio of hard steel balls to powder weight ratio of 4.25:1 at ambient temperature for 48 h in a continuous mode.

lines 110-113: The La1.5Mg0.5Ni7-xMx (Al; x = 0, 0.10, 0.15, 0,20, 0.25 and Mn; x = 0, 0.2, 0.3, 0.4, 0.5) powders synthesized by MA were finally heat-treated in 1123 K for 0.5 h in high purity argon with following cooling in air. For this treatment, the powder (5 g of each composition) was closed under the argon in a quartz tube with a volume of approximately 4 cm3.

- Line 143 & 151: contentment -> content
Response – lines 161, 162: Phase abundance in …

- Lines 145-146: "the Rietveld refinements of La1.5Mg0.5Ni6.85Al0.15 and La1.5Mg0.5Ni6.7Mn0.3 alloys are presented in Fig 1", from the caption of the figure it seems that the Al and Mn content are 0.2 in both alloys.
Response – sorry for our mistake – it should be 0.2 in both cases – see lines 156-157

- Lines 164 and subsequent: did the authors perform any activation procedure for the alloys in order to absorb hydrogen? Which one? Were absorption/desorption cycles performed more than three times (even without measuring the p-c curves but just the hydrogen capacity)?
Response – lines : 124-135

The concentration of the absorbed hydrogen was calculated based on the hydrogen pressure changes measured in the reaction chamber during the tests. The mass of the sample for each measurement cycle was approx. 0.6 g. The investigations of the hydrogen absorption kinetics were carried out at 303 K and under 3 MPa (hydrogen pressure) in the first, second, and third cycle. Each measurement was finished after obtaining the equilibrium hydrogen pressure – the change of pressure did not exceed 200 Pa within 5 min. After each cycle, the samples were degassed at the temperature of 673 K and in a vacuum. The Pressure-Composition-Isotherm (PCI) curves were obtained in the subsequent cycle after the measurements of the kinetics at the same temperature in the hydrogen pressure range up to approx. 7 MPa. The hydrogen absorption and desorption cycles that occurred during the measurements of the kinetics acted as the activation process. The hydrogen content in the samples was obtained by measuring pressures at constant volumes.

- The absorption/desorption curves are not perfectly horizontal. How was the plateau pressure defined? Did the authors consider the middle point of the absorption curves?
Response – the middle point of the absorption curves was used.

- Figure 2: the use of colors to differentiate the curves would help the readers; moreover, the absorption curves extend up to about 70-80 MPa that is 700-800 bar. Are the authors sure? The apparatus they used for those measurements cannot attain these pressure values.
Response – the correct data on new Figs are visible now. See the new Fig 2 in color (page 7).

(a)

(b)

- Table 6 & 7: Define C30 and C50;

Response – see lines 214, 223: *C30 and C50 are the discharge capacity after 30th and 50th charge/discharge cycles, Cmax is the maximum discharge capacity

- Lines 212-213: I do not understand the sentence:"The cycle stability of the La 1.5 Mg 0.5 Ni 7 x M x Al x = 0.10, 0.15, 0.20 and Mn x = 0.2, 0.3 ) electrodes increases." How was stability defined?

Response – lines 214, 223: The stability can be found in Tables 7 and 8 - C30 and C50. For example C50 is the discharge capacity after 50th charge/discharge cycles, Cmax is the maximum discharge capacity.

- Lines 268-272: Is this investigation part of the present work or was it a previous investigation (please, cite).

Response – the mentioned text was removed from the manuscript

We would like to thank you once again for your suggestions for improving our manuscript.

Yours faithfully,

  1. Jurczyk

Round 2

Reviewer 2 Report

The authors followed the suggestions, which improved the clarity of the message they would like to convey and the quality of the manuscript in general. I still have minor recommendations regarding the scientific terminology (please see my comments in the attached PDF file) and this is the only requirement from my side. Once the authors modify the text accordingly, the manuscript can be accepted.

I do not need to see the manuscript again.

Author Response

Dear Reviewer 2,
We are deeply grateful to the suggestions by the reviewer which has enabled us to improve our manuscript. Based on their comments, we have revised our manuscript conscientiously. Changes within the manuscript are marked in red.
The authors followed the suggestions, which improved the clarity of the message they would like to convey and the quality of the manuscript in general. I still have minor recommendations regarding the scientific terminology (please see my comments in the attached PDF file) and this is the only requirement from my side. Once the authors modify the text accordingly, the manuscript can be accepted.
I do not need to see the manuscript again.

We would like to thank you once again for your suggestions for improving our manuscript.

Yours faithfully,
M. Jurczyk

Reviewer 3 Report

Tue Authors addressed the problems posed by the Referee. 

Author Response

Dear Reviewer 3,

We would like to appreciate you giving us the opportunity to resubmit our manuscript: ” Effect of substitutional elements on the thermodynamic and electrochemical properties of mechanically alloyed La1.5Mg0.5Ni7-xMx alloys (M= Al, Mn)” by M. Nowak, M. Balcerzak, M. Jurczyk to Metals.

Tue Authors addressed the problems posed by the Referee. 

We would like to thank you once again for your suggestions for improving our manuscript.

Yours faithfully,

  1. Jurczyk
